# Response of Soybean (*Glycine max* (L.) Merrill) to Mineral Nitrogen Fertilization and *Bradyrhizobium japonicum* Seed Inoculation

**Janusz Prusiński** [1] , **Anna Baturo-Cieśniewska** [2,*] **and Magdalena Borowska** [1]

1   Department of Agronomy, Faculty of Agriculture and Biotechnology, University of Science and Technology, 85-791 Bydgoszcz, Poland; janusz.prusinski@utp.edu.pl (J.P.); borowska@utp.edu.pl (M.B.)

2   Department of Biology and Plant Protection, Faculty of Agriculture and Biotechnology, University of Science and Technology, 85-791 Bydgoszcz, Poland

*   Correspondence: baturo-a@utp.edu.pl; Tel.: +48-52-3749342

**Abstract:** A growing interest in soybean cultivation in Poland has been observed in the recent years, however it faces a lot of difficulties resulting from a poorly understood effectiveness of plant nitrogen fertilization and from the introduction of *Bradyrhizobium japonicum* to the environment. The aim of the study was to evaluate the consistency of response of two soybean cultivars to three different rates of mineral N fertilization and two seed inoculation treatments with *B. japonicum* in field conditions over four years regardless of previous *B. japonicum* presence in the soil. A highly-diversified-over-years rainfall and temperature in the growing season do not allow for a definite statement of the differences resulting from seed inoculation and mineral N fertilization applied separately or jointly in soybean. A high sensitivity of the nodulation process to rainfall deficits was noted, which resulted in a decreased amount of *B. japonicum* DNA measured in qPCR and dry matter of nodules. 'Annushka' demonstrated a higher yield of seeds and protein, higher plants and the 1st pod setting. 'Aldana', due to a significant decrease in plant density, produced a higher number of pods, seeds per pod and the 1000 seed weight per plant. Both cultivars responded with an increase in the seed yield after seed inoculation with HiStick, also with an application of 30 and 60 kg N, as well as with Nitragina with 60 kg N.

**Keywords:** soybean; N fertilization; seed inoculation; yield and its characteristics; *Bradyrhizobium japonicum*

## 1. Introduction

A growing interest in the cultivation of soybean (*Glycine max.* (L.) Merrill.) in Europe and in Poland calls for further research aiming at the determination of the effectiveness of mineral N fertilization and inoculation of soybean seeds with *Rhizobium* bacteria, especially in the fields where soybean has not been cultivated before [1], as well as the effect of the climate variability [2] on the plant yield. In Poland, the average area of soybean stand is only about 20,000 ha [3]; thus, in a vast majority of soils there occur no *Bradyrhizobium japonicum*, a species representing the *Rhizobium* group, as a specific symbiont of this crop [4]. The highly diversified results of research on the effectiveness of inoculating soybean seeds or a simultaneous inoculation and application of mineral nitrogen fertilization in various countries, and, above all, the effectiveness of soybean inoculation have triggered similar studies in Poland. Despite the registration of subsequent cultivars, both Polish and foreign, for many years it has been impossible to significantly increase the soybean acreage in Poland [3].

Most of the N required by soybean plants comes from BNF (biological nitrogen fixation), and the occurrence of bacteria representing the *Rhizobium* group in soil facilitates the plants using from 40–57% [1], 50–60% [5], and up to 50–75% [6] of the symbiotically fixed nitrogen. According to

Osborne and Riedell [7], N fixation increases with the plants' age, reaching the maximum value at R5 stage, i.e., at the beginning of seed formation on one of the four upper nodes of the main shoot. Montanez et al. [8] found that nodulation and the amount of nitrogen derived from atmospheric $N_2$ fixation (Ndfa, nitrogen derived from atmosphere) as well as soybean plant growth are most optimal at the temperature of 25 °C, and significantly worse at the soil temperature of 15 °C. Additionally, a low air temperature can negatively affect the symbiosis of *Rhizobia* bacteria [9], similarly as deficient and excessive rainfall which, at the same time, reduce the soybean yield [10–12]. Therefore, a decreased N content in plants subjected to drought stress indicates a significant decrease in the activity of N fixation [13]. According to Du et al. [14], a long-term water stress, especially if it occurs at the flowering and seed filling stage (BBCH 60–75), decreases the allocation of nutrients to generative parts, thereby reducing the seed weight per pod, as well as the N and protein content in soybean seeds [15]. The ability of soybean plants to feed on mineral N and nitrogen derived from symbiosis with *B. japonicum* is closely related. Brockwell et al. [16] as well as Miransari [17] state that if mineral N is introduced into soil, *B. japonicum* activity, including the amount of N derived from symbiosis, decreases, however, as claimed by Gan et al. [18], no consensus was reached about the quantitative interactions between the supply of mineral N and $N_2$ fixation. According to Tamagno et al. [19], soybean fertilization with mineral N decreases the biological N fixation by 16%, which, according to Saturno et al. [20], results directly from impaired nodulation. Additionally, Kaschuk et al. [21] state that soybean fertilization with mineral N negatively affects not only the number of nodules but also the yield of soybean seeds, while seed inoculation with *B. japonicum* is sufficient to satisfy the nutritional needs of soybean. According to Osborne and Riedell [22], the application of mineral N can increase the plant growth, while, at the same time it reduces nodulation when environmental conditions improve. However, as Popovic et al. [23] state, fertilization with mineral N can reduce the negative drought effects.

The results by Capatana et al. [24] indicate eight-time higher yields of soybean seeds after min. N fertilization than after inoculating seeds with *B. japonicum*. However, according to Albareda et al. [25], mineral N fertilization does not contribute significantly to an increase in soybean yield in comparison with the treatments with inoculation only. These authors indicate a beneficial effect of inoculating seeds with *B. japonicum* on the soybean yield, which, however, gives diversified results. The greatest effects of this treatment were observed in fields without previous soybean cultivation [26]. Cordeiro and Echer [27] found that both mineral fertilization and soil inoculation increase N content in the soil, however the seed yield was clearly higher where the inoculant had been used. The lack of a clearly positive effect of seed inoculation on soybean yield can trigger some doubts as to whether the bacteria introduced proliferated, and if the inoculation was effective, or if environmental conditions and/or nitrogen fertilization did not disturb the development of symbiotic bacteria [7]. Molecular analyses are more and more frequently used for an accurate determination of the degree of *B. japonicum* proliferation. To verify the effectiveness of bacterial inoculations, and thus the number of bacteria in root nodules, quantitative technique real-time PCR (qPCR, quantitative Polymerase Chain Reaction] is used, which is DNA-based and it facilitates, in addition to identifying *B. japonicum*, determining the degree of root colonization by bacteria [28].

Mineral N fertilization enhances the pod number and seed number per pod, however, at rates 25 and 100 kg·ha$^{-1}$ it significantly decreases the number of nodules [29]. As for pure nodulation, mineral fertilization at the rates of 32 kg N·ha$^{-1}$ before sowing + 48 kg N·ha$^{-1}$ has the most beneficial effect on the soybean yield at the stage BBCH 49 [30]. The studies carried out in Poland have shown that soybean fertilization with mineral N increases the seed number per pod and the 1000 seed weight, as compared with the plots with seed inoculation, while the soybean yield significantly depends on the weather pattern in the research years [31]. According to Lorenc-Kozik and Pisulewska [32], N fertilization at the rates of 30 and 60 kg N·ha$^{-1}$ increases the soybean seed yield by 22.7% and 25%, respectively. Jarecki and Bobrecka-Jamro [33] found that inoculation of soybean seeds with HiStick® Soy or Nitragina resulted in an increase in the number and dry weight of nodules, the pod number per plant as well as in SPAD (leaf greenness index) and LAI (leaf area index).

We still do not have enough information about the presence and activity of *B. japonicum* in Polish soils. The knowledge on the relationship between inoculation and N mineral fertilization at various rates is still insufficient as those factors have been only partially tested by Jarecki and Bobrecka-Jamro [31,33]. In addition, a large variation in humidity and temperature has been observed for several recent years in Poland, so it is reasonable to verify the reaction of soybean to both factors under such conditions. It is important primarily from an economic point of view as well as determining the applicability of the treatments. The research hypothesis assumes a high efficiency of soybean seed inoculation to reduce the use of mineral N for environmental and financial reasons. The aim of our research has been to assess the effect of mineral N fertilization and soybean seed inoculating with *B. japonicum* on the yield, content and protein yield and some traits, e.g., number of pods per plant and seeds per pod, also the weight of seeds per pod, the 1000 seed weight as well as plant height, first pod height, harvest index, LAI, and the content of *B. japonicum* DNA in roots of soybean over 4 years different in terms of temperature and humidity.

## 2. Materials and Methods

### 2.1. Site Description

The three-factor experiments of (I) two soybean cultivars, (II) the method of nitrogen supply/uptake (N fertilization and/or seed inoculation) and (III) year with *B. japonicum* together or without N fertilization were set up in 2016–2019 with spring wheat as the forecrop, at the Research Station of the UTP University of Science and Technology at Mochełek (latitude 53°13′, longitude 17°51′, 89.8 mamsl), Poland. The soil researched was a typical lessive soil from light loamy sands, deposited in a shallow layer on light loam, Haplic Luvisols (Cutanic) [34]. The mean mineral N content over the research years in the profile 0–60 cm was 59.8 kg·ha$^{-1}$, and with every research year it was significantly reduced. The content of $P_2O_5$ was very high (20.8 mg·100 g$^{-1}$ of soil), the potassium content was high (16.2 mg·100 g$^{-1}$ of soil), while the magnesium content was low (2.83 mg·100 g$^{-1}$ of soil). In all research years, the soil pH was suitable for soybean cultivation (Table 1).

**Table 1.** Chemical properties of soil at the depth of 0–60 cm.

| Years | mg·100 g$^{-1}$ of Soil | | | mg·kg$^{-1}$ of Soil | | N min. kg ha$^{-1}$ | pH in KCl |
|---|---|---|---|---|---|---|---|
| | $P_2O_5$ | $K_2O$ | Mg | N-NO$_3$ | N-NH$_4$ | | |
| 2016 | 25.0 | 15.3 | 3.0 | 9.53 | 5.73 | 68.5 | 6.3 |
| 2017 | 21.9 | 18.0 | 3.1 | 5.90 | 7.96 | 62.5 | 6.3 |
| 2018 | 18.8 | 13.5 | 2.7 | 6.12 | 6.97 | 58.9 | 6.8 |
| 2019 | 17.6 | 18.0 | 2.7 | 8.69 | 2.27 | 49.3 | 6.5 |
| Mean | 20.8 | 16.2 | 2.87 | 7.56 | 5.73 | 59.8 | 6.5 |

### 2.2. Experimental Design

The research involved the earliest cultivars suitable for cultivation in Poland: Aldana (000) early cultivar from the Polish breeding; and a very early one from the Ukrainian breeding, cv. Annushka (0000). Both cultivars were fertilized with mineral N (0, 30 and 60 kg·ha$^{-1}$) before planting, and/or their seed material was inoculated with two inoculants—domestic Nitragina (IUNG-PIB, PL), or foreign HiStick® Soy (BASF Agricultural Specialities Limited, GB), separately and in combination with mineral N fertilization. Ammonium nitrate—salt of nitric acid and ammonia (NH$_4$NO$_3$) was used. The salt was applied on the soil surface and mixed with soil before sowing. Both Nitragina and HiStick® Soy contain *B. japonicum* strains specific for soybean, in which peat is a carrier.

### 2.3. Data Collection

The plots for sowing were 21.24 m$^2$ in size and for harvest—20.0 m$^2$ (Wintersteiger classic plot combine), row spacing—16 cm, sowing rate—90 germinating seeds per m$^2$, and the sowing depth 3–4 cm. Both cultivars were sown in subsequent research years on 25 April, 26 April, 25 April, and

on 6 May, respectively. Every year, before sowing 60 kg P and 80 kg·ha$^{-1}$ K were used. Directly after sowing, Sencor (550 mL·ha$^{-1}$) was applied into soil, and after emergence, Corum 502.4 SL with adjuvant Dash HC (1.25 + 1.0 dm$^3$·ha$^{-1}$), as well as Fusilade Forte (800 mL·ha$^{-1}$) to control annual monocotyledonous weeds. To control *Vanessa cardui* infestation in 2019, Proteus was applied at a rate of 750 mL·ha$^{-1}$. At the full flowering stage R2 [35] LAI (Leaf Area Index, m$^2$/m$^2$) was determined in soybean plants with the use of SunScan Canopy Analysis System (ΔT Devices Ltd., Cambridge, UK) in 4 replications. The plant nutrition was assayed with a chlorophyll meter SPAD 502P (Konica Minolta Europe). The measurements were taken for the leaves of 30 plants in each combination, in four replications. The N content (nitrogen %) in the dry weight of seeds (DWS) was determined with the Kjeldahl method in accordance with PN-75/A-04018 [36].

In 2017–2019, molecular analyses were performed with the use of Real-time PCR technique, aiming at quantitative analysis of *B. japonicum* in DNA samples of cv. Aldana, extracted from the roots. For this purpose, at stage R5, 10 roots were randomly collected from each plot in 3 replications from 9 experimental combinations. The roots were thoroughly cleaned of soil, lyophilized (CoolSAFE, Scanvac, Denmark) and homogenized in a ball mill (Retsch MM400, Haan, Germany). The DNA was extracted from 50 mg of the fine powder following the modified Doyle and Doyle method [37] in 900 μL of extraction buffer, which contained CTAB, EDTA, NaCl, Tris-HCl, β-mercaptoethanol and PVP. After incubation at the temperature of 65 °C, to remove proteins and carbohydrates, phenol, chloroform and isoamyl alcohol were used. DNA was precipitated with 95% ethyl alcohol and suspended in ddH$_2$O. The DNA was measured fluorometrically (Quantus, Promega, WI, USA), and for further analyses, it was diluted to the concentration of 1 ng·μL$^{-1}$ in ddH$_2$O. The qPCR analyses were performed for each of the 27 samples in three replications in a reaction mixture consisting of: 5 μL LightCycler480 SYBR Green I Master (Roche, Basel, Switzerland); 0.25 μL of each starter (10 pM·μL$^{-1}$) nodZ-A-F (5'GGTTTGGCGACTGTCTGTGGTC-3'), as well as nodZ-A-R (5'-TTCCACCATGTTGGAAAGAATGGTCC-3') [28], and 4.5 μL DNA under conditions (of): preincubation 10 min at 95 °C and 40 cycles of amplification, which consisted of: denaturation at 95 °C for 10s, hybridization at 64 °C for 20 s, as well as extension at 72 °C for 30 s. To verify the specificity of the reactions, a melting curve was developed. The standard in qPCR analysis was DNA of the reference isolate of *B. japonicum* NCCB 47038, based on which also the standard curve was developed. The quantity of *B. japonicum* pg DNA in the total DNA in samples subjected to quantitative analysis was calculated with the use of the standard curve and software package LightCycler (Roche). After reaching full maturity, the seed yield was determined, as well as the most important morphological traits of plants and structural components of soybean yield.

## 2.4. Statistical Analysis

The results of the three-factor field and laboratory experiment (I—cultivar, II—N fertilization and/or seed inoculation, III—year), as well as the results of molecular analyses, were subjected to the ANOVA in a randomized complete block design with four replications. Means ± standard error was performed using Fisher's least significant difference (LSD) at $p \leq 0.05$. In the case of multi factor analyses, a Tukey's honestly significant difference (HSD) test was used. For analysis of long-term observations, a repeated measures ANOVA was used. Pearson's correlation coefficient was calculated to quantify the strengths of the relationship between the contents of DNA of *B. japonicum* and dry nodules weight and between yield and rainfall. The data of each variable in this analysis includes the mean of 10 root samples from each plot in 2017–2019 (*n* = 81).

Statistical analyses of the seed yield and morphological and biological traits of plants and seeds were made using software Statistica$^®$ 13.3. The means in the tables and charts denoted by the same letters did not differ significantly at * $p \leq 0.05$ or ** $p \leq 0.01$.

## 3. Results

Plant growth in 2018 and 2019 was accompanied by the mean monthly air temperature of 20.2 and 19.1 °C in July and August, i.e., higher than in 2016 and 2017 (17.3 and 17.7 °C) (Figure 1). On the other hand, a high and well-distributed total rainfall in July and August was recorded only in 2016 and 2017 (98.1 and 133.8 mm, as well as 119 and 126 mm, respectively), which facilitated a good growth of plants, their development and yield. However, in subsequent research years a soil drought occurred; in 2018 merely 86 mm of rainfall was observed in July and 23.7 mm in August, while in 2019—22.4 mm and 37.7 mm, respectively. Rainfall in September in those years, no longer affected the soybean yield, while pods and seeds dried slowly, substantially extending the growing season in both cultivars up to about 140 days.

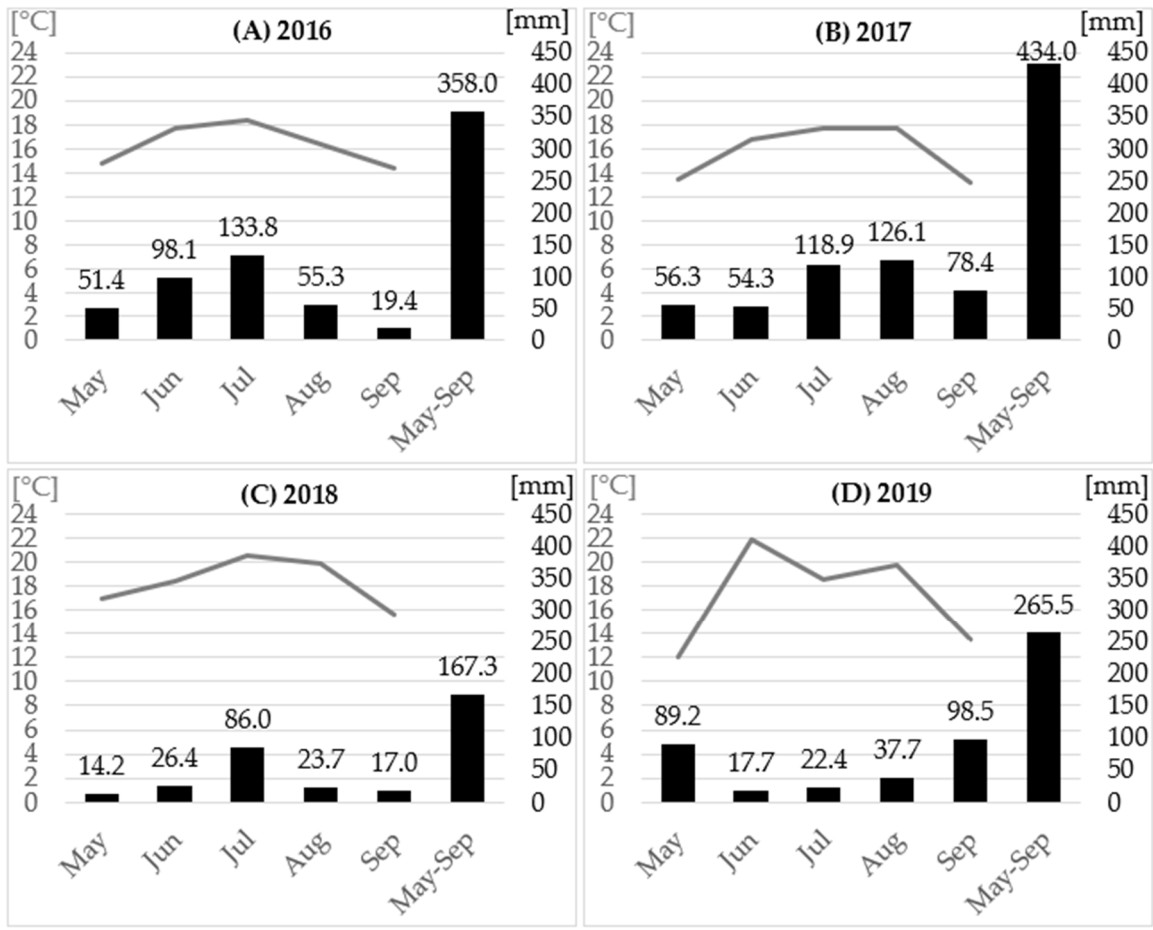

**Figure 1.** Mean monthly air temperature (°C) and total rainfall (mm) in (**A**) 2016; (**B**) 2017; (**C**) 2018 and (**D**) 2019.

Only in 2016 and 2017, a higher yield of cv. Annushka was observed with a considerable diversification in the seed yield in both cultivars, resulting from mineral N fertilization and/or seed inoculation (Table 2). In the highly favorable, in terms of the total rainfall, year 2016, statistically similar seed yields in both soybean cultivars were harvested after inoculating seeds with HiStick, and also in combination of HiStick with 30 and 60 kg N·ha$^{-1}$, in 2017—also with Nitragina + 60 kg N·ha$^{-1}$. In other years, almost no significant differences were observed in the yield of both cultivars, including those resulting from the methods of inoculation and/or mineral N fertilization applied. In the four years of research, very high differences were found in the yield of both soybean cultivars resulting mainly from a changeable humidity over the research years, and a significant difference in the plant density between both cultivars.

**Table 2.** Seed yield of cv. Aldana (AL) and Annushka (AN) depending on the inoculation and mineral N fertilization applied in 2016; 2017; 2018 and 2019 (t·ha$^{-1}$).

| Year (Y) | 2016 | | 2017 | | 2018 | | 2019 | | 2016–2019 | |
|---|---|---|---|---|---|---|---|---|---|---|
| Cultivar (C) | AL | AN | AL | AN | AL | AN | AL | AN | AL | AN |
| Treatment (T) | | | | | | | | | | |
| Control | 2.58 ± 0.2c | 3.72 ± 0.25d | 2.2 ± 0.5 | 3.18 ± 0.17d | 1.58 ± 0.11 | 2.07 ± 0.25 | 1.00 ± 0.07 | 1.21 ± 0.06 | 1.84 ± 0.2cd | 2.54 ± 0.27c |
| 30 kg N·ha$^{-1}$ | 2.5 ± 0.21c | 3.54 ± 0.16d | 2.00 ± 0.13 | 3.34 ± 0.1d | 1.72 ± 0.16 | 1.87 ± 0.15 | 0.91 ± 0.05 | 1.44 ± 0.24 | 1.78 ± 0.16d | 2.55 ± 0.25c |
| 60 kg N·ha$^{-1}$ | 3.23 ± 0.18b | 4.18 ± 0.14abc | 2.26 ± 0.34 | 3.51 ± 0.2cd | 1.85 ± 0.12 | 1.61 ± 0.27 | 0.99 ± 0.05 | 1.26 ± 0.14 | 2.08 ± 0.23bcd | 2.64 ± 0.33c |
| HiStick0 kg N·ha$^{-1}$ | 3.93 ± 0.13a | 4.69 ± 0.41a | 2.42 ± 0.4 | 3.99 ± 0.07ab | 1.84 ± 0.16 | 1.86 ± 0.21 | 0.86 ± 0.07 | 1.6 ± 0.09 | 2.26 ± 0.3ab | 3.03 ± 0.36a |
| Nitragina0 kg N·ha$^{-1}$ | 2.64 ± 0.08c | 3.61 ± 0.22d | 2.66 ± 0.1 | 3.81 ± 0.06bc | 1.82 ± 0.18 | 1.8 ± 0.14 | 0.84 ± 0.07 | 1.47 ± 0.09 | 1.99 ± 0.2bcd | 2.67 ± 0.28bc |
| HiStick + 30 kg N·ha$^{-1}$ | 3.86 ± 0.17a | 4.38 ± 0.56ab | 2.49 ± 0.37 | 4.19 ± 0.15ab | 1.87 ± 0.16 | 1.76 ± 0.19 | 0.85 ± 0.05 | 1.41 ± 0.08 | 2.27 ± 0.3ab | 2.94 ± 0.38ab |
| HiStick + 60 kg N·ha$^{-1}$ | 3.96 ± 0.06a | 4.53 ± 0.16a | 2.75 ± 0.39 | 4.34 ± 0.13a | 1.8 ± 0.11 | 1.72 ± 0.1 | 1.18 ± 0.18 | 1.69 ± 0.1 | 2.42 ± 0.29a | 3.07 ± 0.36a |
| Nitragina + 30 kg N·ha$^{-1}$ | 2.97 ± 0.2bc | 3.49 ± 0.22cd | 2.66 ± 0.25 | 3.89 ± 0.13bc | 1.83 ± 0.18 | 1.8 ± 0.2 | 0.84 ± 0.09 | 1.33 ± 0.07 | 2.07 ± 0.23bcd | 2.63 ± 0.29c |
| Nitragina + 60 kg N·ha$^{-1}$ | 3.36 ± 0.18b | 3.97 ± 0.35bcd | 2.28 ± 0.53 | 3.96 ± 0.25ab | 1.88 ± 0.2 | 1.83 ± 0.15 | 0.94 ± 0.04 | 1.4 ± 0.1 | 2.12 ± 0.26abc | 2.79 ± 0.32abc |
| Mean | 3.23 ± 0.16b | 4.01 ± 0.27a | 2.42 ± 0.3b | 3.8 ± 0.14a | 1.8 ± 0.2 | 1.81 ± 0.18 | 0.93 ± 0.07b | 1.42 ± 0.11a | 2.09 ± 0.08b | 2.76 ± 0.1a |
| | 3.62 ± 0.09a | | 3.11 ± 0.11b | | 1.8 ± 0.04c | | 1.18 ± 0.04d | | | |

| ANOVA | | | | | | | | | | |
|---|---|---|---|---|---|---|---|---|---|---|
| | 2016 | | 2017 | | 2018 | | 2019 | | 2016–2019 | |
| | AL | AN | AL | AN | AL | AN | AL | AN | | |
| T | ** | * | n.s. | ** | n.s. | n.s. | n.s. | n.s. | ** | |
| C | ** | | ** | | n.s. | | ** | | ** | |
| Y | i.m. | | i.m. | | i.m. | | i.m. | | ** | |
| C × T | n.s. | | n.s. | | n.s. | | n.s. | | n.s. | |
| C × T × Y | i.m. | | i.m. | | i.m. | | i.m. | | n.s. | |

Mean ± standard error followed by different letters in the same cultivar in each year are significantly different at $p \leq 0.05$ according to Tukey's honestly significant difference (HSD) test. * Significant at $p \leq 0.05$; ** significant at $p \leq 0.01$; n.s.—non-significant, i.m.—impossible to measure.

The average long-term seed yield of both soybean cultivars was 2.30 t·ha$^{-1}$. Almost in every subsequent research year, the seed yield in both cultivars was reduced. The seed yield of cv. Annushka was 28.6% higher than in cv. Aldana.

The correlation analysis showed a significant relationship between the sum of rainfall in June–August and the seed yield of soybean seeds (Table 3). With the increase in rainfall by 0.36 mm, the yield increased by 0.012 t·ha$^{-1}$. A very strong correlation ($r = 0.99$) has been found here, and the described model explains 98% of the studied variable. Strong relationship between rainfall at the stage of flowering and pod filling was also noticed in individual months: June, July, August, and the entire cropping season, but it could not be statistically confirmed. In our experiment, rainfall in May and September did not correlate with the obtained yield, which, however, was not statistically proven.

**Table 3.** Pearson correlation coefficients (r), and coefficient of determination ($R^2$) for rainfall sum in different months and seed yield.

| Month | Linear Function | *r* Pearson Value | $R^2$ | *p*-Value |
|---|---|---|---|---|
| May | y = 2.9506 − 0.0036 × x | −0.08 | 0.01 | 0.92 |
| Jun | y = 1.131 + 0.0332 × x | 0.90 | 0.81 | 0.10 |
| Jul | y = 0.5559 + 0.0244 × x | 0.91 | 0.82 | 0.10 |
| Aug | y = 1.5152 + 0.0205 × x | 0.70 | 0.49 | 0.30 |
| Sep | y = 3.1994 − 0.0082 × x | −0.26 | 0.07 | 0.74 |
| Jun-Aug | y = 0.3645 + 0.012 × x | 0.99 | 0.98 | **0.01** |
| May-Sep | y = −0.1832 + 0.0096 × x | 0.83 | 0.69 | 0.17 |

Values in bold are considered to be significant.

To sum up, in the long-term period, both cultivars yielded higher after the application of seed inoculation with HiStick, HiStick + 30 kg N·ha$^{-1}$, HiStick + 60 kg N·ha$^{-1}$ and Nitragina + 60 kg N·ha$^{-1}$ than with other treatments applied. Mineral N fertilization at the rates of 30 and 60 kg N·ha$^{-1}$, and also the application of Nitragina alone, or with an addition of 30 kg N·ha$^{-1}$ in both cultivars, did not contribute to a significant increase in the seed yield in both cultivars.

The average long-term protein content was 36.1% and protein yield was 816 kg·ha$^{-1}$ (Table 4). Similar to the seed yield, also the protein yield in both cultivars was reduced over years and in cv. Annushka it was 27.3% higher than in cv. Aldana. Differences in the protein content in seeds in both cultivars were found. The highest protein content in seeds of cv. Annushka was recorded after the use of HiStick + 60 kg N·ha$^{-1}$, and significantly lower for the control, 30 kg N·ha$^{-1}$, and Nitragina without N as well as with both rates of N. The protein content in seeds of the cv. Aldana was significantly higher after the application of HiStick without N as well as with both rates of N. The protein yield of the cv. Annushka was the highest when using HiStick alone and HiStick with both rates of N, compared to other combinations. For cv. Aldana the highest protein yield was recorded in the case of HiStick and HiStick + 60 kg N·ha$^{-1}$, but these values were not significantly different from the combinations where 60 kg N·ha$^{-1}$, Nitragina without N, HiStick + 30 kg N·ha$^{-1}$ and Nitragina + 60 kg N·ha$^{-1}$ were applied.

Thermal conditions for soybean cultivation in Poland, especially at the germination and emergence stage in 2017 and 2019 were highly unfavorable (low temperature), especially for cv. Aldana, and contributed to a reduction in the average plant density per m$^2$ in 2016–2019 to 38.7, whereas the average plant density in cv. Annushka was 71.3. The cultivars differed in the majority of traits conditioning the soybean yield (Table 5). Cv. Aldana was showed a higher number of pods, a higher seed weight per pod, and a higher 1000 seed weight than cv. Annushka. Only the seed number per pod in the cultivars studied did not differ. The seed weight per pod in cv. Aldana was higher than in cv. Annushka after applying 60 kg N, preparation HiStick, HS + 30 and + 60 kg N, as well as after applying Nitragina at the rate of 30 and 60 kg N.

**Table 4.** Protein content and seed protein yield in soybean cv. Aldana and Annushka.

| Research Years | Protein Content in Seeds (%) | | Protein Yield (kg·ha⁻¹) | |
|---|---|---|---|---|
| | Aldana | Annushka | Aldana | Annushka |
| 2016 | 35.4a | 34.3a | 1130a | 1406a |
| 2017 | 38.9a | 37.2a | 665b | 1346a |
| 2018 | 35.9a | 34.0a | 708b | 661b |
| 2019 | 38.1a | 35.9a | 352c | 467c |
| Treatment | | | | |
| Control | 35.9b | 34.4c | 598d | 823c |
| 30 kg N·ha⁻¹ | 36.3b | 35.1bc | 623cd | 866bc |
| 60 kg N·ha⁻¹ | 37b | 35.9ab | 784abc | 918bc |
| HiStick, 0 kg N·ha⁻¹ | 38.7a | 36.1ab | 845a | 1112a |
| Nitragina, 0 kg N·ha⁻¹ | 36.7b | 34.4c | 670abcd | 925bc |
| HiStick + 30 kg N·ha⁻¹ | 39a | 35.9abc | 816ab | 1154a |
| HiStick + 60 kg N·ha⁻¹ | 39.5a | 37.3a | 835a | 1138a |
| Nitragina + 30 kg N·ha⁻¹ | 36.7b | 35.0bc | 640bcd | 921bc |
| Nitragina + 60 kg N·ha⁻¹ | 37.2b | 35.7bc | 703abcd | 965b |
| Mean for cultivars and years | 37.0a | 35.3b | 714b | 970a |
| **Mean** | 36.1 | | 816 | |
| ANOVA | | | | |
| | Protein content in seeds (%) | | Protein yield (kg·ha⁻¹) | |
| **C** | ** | | ** | |
| **T** | ** | | ** | |
| **Y** | ** | | ** | |
| **C × T** | n.s | | n.s | |
| **C × T × Y** | n.s | | n.s | |

Means followed by different letters are significantly different at $p \leq 0.05$. According to the LSD test. ** significant at $p \leq 0.01$; n.s.—non-significant.

**Table 5.** Effect of the mineral fertilization and inoculants applied on the essential morphological traits in soybean.

| Treatment (T) | Pod Number per Plant | Seed Number per Pod | Weight of Seeds per Pod (g) | Weight of 1000 Seeds (g) |
|---|---|---|---|---|
| Aldana | | | | |
| Control | 34.45 ± 5.52 | 1.75 ± 0.06b | 0.32 ± 0.01c | 185.68 ± 4.36 |
| 30 kg N·ha⁻¹ | 37.01 ± 7.08 | 1.75 ± 0.05b | 0.33 ± 0.01bc | 188.26 ± 5.26 |
| 60 kg N·ha⁻¹ | 31.96 ± 5.13 | 1.72 ± 0.07b | 0.33 ± 0.01bc | 193.52 ± 2.38 |
| HiStick, 0 kg N·ha⁻¹ | 35.51 ± 5.54 | 1.92 ± 0.08a | 0.39 ± 0.02a | 202.34 ± 7.7 |
| Nitragina, 0 kg N·ha⁻¹ | 37.05 ± 6.68 | 1.75 ± 0.06b | 0.33 ± 0.01c | 188.02 ± 4.12 |
| HiStick + 30 kg N·ha⁻¹ | 33.72 ± 5.88 | 1.73 ± 0.09b | 0.35 ± 0.02bc | 202.05 ± 4.06 |
| HiStick + 60 kg N·ha⁻¹ | 36.73 ± 6.94 | 1.77 ± 0.06b | 0.36 ± 0.02ab | 204.18 ± 3.4 |
| Nitragina + 30 kg N·ha⁻¹ | 35.13 ± 5.8 | 1.7 ± 0.07b | 0.33 ± 0.01bc | 197.56 ± 9.16 |
| Nitragina + 60 kg N·ha⁻¹ | 32.16 ± 4.64 | 1.75 ± 0.07b | 0.33 ± 0.02bc | 190.41 ± 3.23 |
| Mean | 34.86 ± 1.94a | 1.76 ± 0.02 | 0.34 ± 0.01a | 194.67 ± 1.8a |
| Annushka | | | | |
| Control | 17.2 ± 1.85 | 1.91 ± 0.05a | 0.27 ± 0.01 | 143.06 ± 3.08 |
| 30 kg N·ha⁻¹ | 18.32 ± 1.77 | 1.82 ± 0.05ab | 0.27 ± 0.01 | 146.65 ± 4.21 |
| 60 kg N·ha⁻¹ | 17.76 ± 1.76 | 1.72 ± 0.05b | 0.25 ± 0.01 | 147.21 ± 4.55 |
| HiStick, 0 kg N·ha⁻¹ | 19.3 ± 1.99 | 1.83 ± 0.06ab | 0.27 ± 0.01 | 147.75 ± 4.38 |
| Nitragina, 0 kg N·ha⁻¹ | 15.85 ± 1.32 | 1.84 ± 0.05ab | 0.27 ± 0.01 | 150.01 ± 3.88 |
| HiStick + 30 kg N·ha⁻¹ | 17.6 ± 1.93 | 1.83 ± 0.05ab | 0.28 ± 0.01 | 154.48 ± 3.21 |
| HiStick + 60 kg N·ha⁻¹ | 20.53 ± 1.91 | 1.79 ± 0.06ab | 0.28 ± 0.01 | 153.51 ± 4.17 |
| Nitragina + 30 kg N·ha⁻¹ | 17.98 ± 1.69 | 1.75 ± 0.07b | 0.27 ± 0.01 | 155.69 ± 7.47 |
| Nitragina + 60 kg N·ha⁻¹ | 18.26 ± 1.58 | 1.71 ± 0.06b | 0.26 ± 0.01 | 152.36 ± 5.57 |
| **Mean** | 18.09 ± 0.58b | 1.8 ± 0.02 | 0.27 ± 0.00b | 150.08 ± 1.55b |
| ANOVA | | | | |
| **C** | ** | n.s. | ** | ** |
| **T** | n.s. | * | ** | n.s. |
| **C × T** | n.s. | n.s. | n.s. | n.s. |

Mean ± standard error followed by different letters in the same cultivar is significantly different at $p \leq 0.05$. according to the LSD test. * Significant at $p \leq 0.05$; ** significant at $p \leq 0.01$; n.s.—non-significant.

On the other hand, the plants of cv. Annushka demonstrated an increased plant height and the height of the 1st pod setting as compared with cv. Aldana (Table 6). Its plant height was most enhanced by fertilization with HiStick + 60 kg N·ha⁻¹, although the other treatments applied, except the control and Nitragina, did not differentiate the plant height. The Harvest index (HI) value in cv. Aldana was higher than in Annushka only for the control plot, and after using Nitragina + 60 kg N·ha⁻¹; in the other combinations no such significant variation was observed in both cultivars. Additionally, only when applying HiStick as well as HiStick + 60 kg N·ha⁻¹, a higher LAI index value was found in cv. Annushka.

**Table 6.** Plant height and height of the 1st pod setting, harvest index and leaf area index (LAI) in the soybean cultivars studied in 2016–2019.

| Treatment (T) | Plant Height (cm) | First Pod Height(cm) | Harvest Index | LAI |
|---|---|---|---|---|
| Aldana | | | | |
| Control | 54.94 ± 4.73 | 6.26 ± 0.79 | 0.5 ± 0.02 | 2.3 ± 0.33b |
| 30 kg N·ha$^{-1}$ | 51.51 ± 4.03 | 6.45 ± 0.74 | 0.48 ± 0.02 | 3.17 ± 0.53a |
| 60 kg N·ha$^{-1}$ | 54.16 ± 5.3 | 6.72 ± 0.72 | 0.49 ± 0.02 | 3.4 ± 0.57a |
| HiStick, 0 kg N·ha$^{-1}$ | 52 ± 5.17 | 6.48 ± 0.85 | 0.51 ± 0.03 | 2.7 ± 0.28ab |
| Nitragina, 0 kg N·ha$^{-1}$ | 51.74 ± 5.14 | 5.73 ± 0.85 | 0.48 ± 0.02 | 2.5 ± 0.31ab |
| HiStick + 30 kg N·ha$^{-1}$ | 52.84 ± 4.95 | 6.48 ± 0.79 | 0.47 ± 0.03 | 3.03 ± 0.52a |
| HiStick + 60 kg N·ha$^{-1}$ | 52.5 ± 4.93 | 6.77 ± 1.08 | 0.48 ± 0.02 | 2.82 ± 0.39ab |
| Nitragina + 30 kg N·ha$^{-1}$ | 53.06 ± 4.73 | 6.26 ± 0.87 | 0.49 ± 0.02 | 2.82 ± 0.38ab |
| Nitragina + 60 kg N·ha$^{-1}$ | 52.16 ± 5.47 | 5.86 ± 0.89 | 0.52 ± 0.01 | 3.05 ± 0.58a |
| Mean | 52.77 ± 1.61b | 6.33 ± 0.28b | 0.49 ± 0.01 | 2.86 ± 0.14b |
| Annushka | | | | |
| Control | 64.49 ± 5.24c | 12.49 ± 1.23a | 0.46 ± 0.02 | 3.58 ± 0.77b |
| 30 kg N·ha$^{-1}$ | 73.78 ± 8.06ab | 11.14 ± 1b | 0.49 ± 0.02 | 3.2 ± 0.99b |
| 60 kg N·ha$^{-1}$ | 73.79 ± 8.24ab | 11.19 ± 1.02b | 0.48 ± 0.03 | 4.6 ± 0.78a |
| HiStick, 0 kg N·ha$^{-1}$ | 69.85 ± 8.05abc | 10.76 ± 1.1b | 0.5 ± 0.01 | 4.23 ± 0.83ab |
| Nitragina, 0 kg N·ha$^{-1}$ | 65.79 ± 6.8c | 12.41 ± 1.02a | 0.52 ± 0.01 | 3.88 ± 0.82ab |
| HiStick + 30 kg N·ha$^{-1}$ | 70.81 ± 7.94abc | 12.66 ± 1.17a | 0.51 ± 0.02 | 4.47 ± 0.77a |
| HiStick + 60 kg N·ha$^{-1}$ | 76 ± 8.48a | 10.71 ± 0.76b | 0.47 ± 0.02 | 4.83 ± 0.85a |
| Nitragina + 30 kg N·ha$^{-1}$ | 70.41 ± 7.53abc | 11.74 ± 1.26ab | 0.5 ± 0.01 | 4.25 ± 0.78ab |
| Nitragina + 60 kg N·ha$^{-1}$ | 70.03 ± 7.32abc | 10.94 ± 0.99b | 0.47 ± 0.02 | 4.48 ± 1.06a |
| **Mean** | 70.55 ± 2.47a | 11.56 ± 0.35a | 0.49 ± 0.01 | 4.17 ± 0.27a |
| ANOVA | | | | |
| **C** | ** | ** | n.s. | ** |
| **T** | ** | n.s. | n.s. | * |
| **C × T** | * | * | ** | n.s. |

Mean ± standard error followed by different letters in the same cultivar is significantly different at $p \leq 0.05$. According to the Tukey's HSD test. * Significant at $p \leq 0.05$; ** significant at $p \leq 0.01$; n.s.—non-significant.

The research on *B. japonicum* DNA started in 2017 and indicated most DNA of the symbiont under a favorable humidity in this year **(Table 7)**. Its amount was almost 4.8 times higher than in 2018, and up to 40 times higher than in 2019. The differences in the amount of pg DNA in 2018 and 2019 were lower, however, also significantly diversified—in the very dry year 2019 the amount of DNA was lower than in 2018. The same linear response of cv. Aldana to a progressively less favorable humidity over the years was recorded for the dry weight of nodules (DNW).

**Table 7.** *B. japonicum* DNA in roots (pg) and dry nodule weight (DNW) (g) per plant over the research years.

| Research Years | DNA *B. japonicum* (pg) | DNW (g) |
|---|---|---|
| 2017 | 24301.63 ± 3641.22a | 0.46 ± 0.06a |
| 2018 | 4009.05 ± 474.10b | 0.29 ± 0.04b |
| 2019 | 534.45 ± 185.04c | 0.12 ± 0.02c |
| Mean | 9615.04 ± 1684.53 | 0.29 ± 0.03 |
| ANOVA | | |
| **Y** | ** | ** |

Mean ± standard error followed by different letters is significantly different at $p \leq 0.05$. According to the LSD test. ** significant at $p \leq 0.01$.

Significant differences in the amount of *B. japonicum* DNA resulted from the diversified total rainfall in subsequent research years (Table 8). Importantly, when exposed to a favorable humidity in 2017, there was recorded a positive response to the seed inoculation; the highest and significantly similar amount of DNA was found after the inoculation of soybean seeds with HiStick, HiStick + 30 kg N·ha$^{-1}$ and Nitragina. In the other two dry years, seed inoculation, fertilization, or their interaction, did not affect the amount of the bacterial DNA. On average, in the long-term period of 2017–2019 in particular experimental combinations, significant differences were indicated in the amount of *B. japonicum* DNA. More DNA was observed in 5 combinations, where HiStick (17,087.2 pg) and Nitragina (15,149.3 pg)

were used for inoculation with a simultaneous inoculation and fertilization at the rate of 30 kg N·ha$^{-1}$ (HiStick + 30 N·ha$^{-1}$—14,446.1 pg and Nitragina + 30 N·ha$^{-1}$—12,188.0 pg), and also when applying HiStick + 60 kg N·ha$^{-1}$ (12,690.2 pg), than in the combinations where seeds were not inoculated with any preparations containing *B. japonicum* (control—24.79 pg, 30 N·ha$^{-1}$—3690.7 pg, 60 N·ha$^{-1}$—1269.6 pg) and where Nitragina was applied with a simultaneous nitrogen fertilization at the rate of 60 kg·ha$^{-1}$ (2892.1 pg).

**Table 8.** Effect of the applied treatments on the content of *B. japonicum* DNA in roots over the research years.

| Treatment | Year | | | 2017–2019 |
|---|---|---|---|---|
| | 2017 | 2018 | 2019 | |
| | DNA *B. japonicum* (pg) | | | |
| Control | 62.93 ± 5.86b | 9.57 ± 1.18c | 2.40 ± 1.12c | 24.97 ± 9.71b |
| 30 kg N·ha$^{-1}$ | 5073.33 ± 1393.35b | 5996.33 ± 198.32a | 2.43 ± 0.3c | 3690.7 ± 1016.37b |
| 60 kg N·ha$^{-1}$ | 816.20 ± 387.11b | 2990 ± 1031.72b | 2.60 ± 1.19c | 1269.6 ± 547.69b |
| HiStick, 0 kg N·ha$^{-1}$ | 44112 ± 8027.49a | 6351.00 ± 263.85a | 798.80 ± 161.17b | 17087.27 ± 7187.92a |
| Nitragina, 0 kg N·ha$^{-1}$ | 38233.33 ± 12268.18a | 4492 ± 410.43a | 2722.67 ± 580.53a | 15149.33 ± 6778.95a |
| HiStick + 30 kg N·ha$^{-1}$ | 37073 ± 3583.15a | 5608 ± 444.05a | 657.43 ± 170b | 14446.14 ± 5796.36a |
| HiStick + 60 kg N·ha$^{-1}$ | 31989 ± 4623.99a | 6073.33 ± 408.83a | 8.40 ± 1.73c | 12690.24 ± 5083.27a |
| Nitragina + 30 kg N·ha$^{-1}$ | 31900 ± 3342.15a | 4161.67 ± 823.68a | 502.53 ± 204.89b | 12188.07 ± 5055.18a |
| Nitragina + 60 kg N·ha$^{-1}$ | 4022 ± 1094.08b | 4596.67 ± 427.39a | 57.83 ± 18.79c | 2892.17 ± 789.92b |
| ANOVA | | | | |
| Degree of freedom | 8 | 8 | 8 | 8 |
| F–test | 11.58 | 13.99 | 16.13 | 2.21 |
| *p*–value | <0.001 | <0.001 | <0.001 | 0.037 |

Mean ± standard error followed by different letters is significantly different at $p \leq 0.05$, according to the LSD test.

Additionally, a high correlation ($r = 0.68$) was observed between the amount of *B. japonicum* DNA and the dry weight of soybean nodules (Figure 2). The increasing content of the dry weight of nodules was accompanied by an increase in the DNA quantity, although also numerous single deviations from that finding were observed.

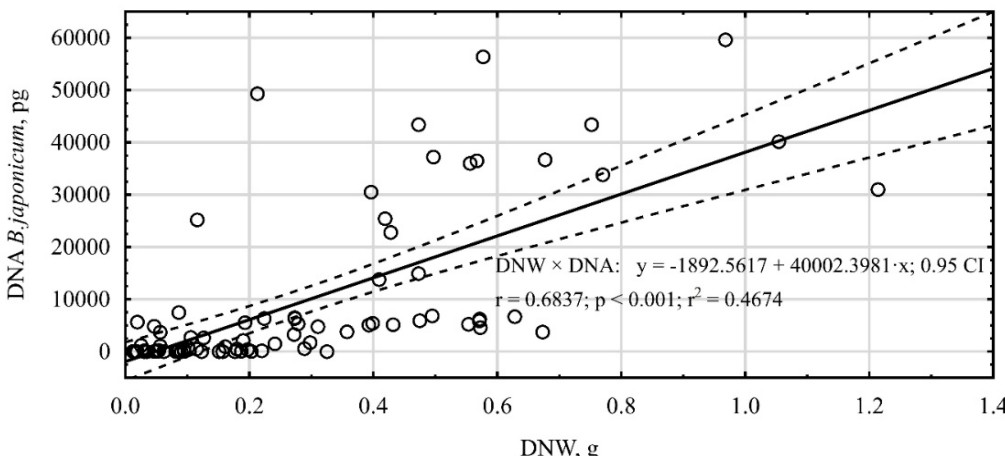

**Figure 2.** Correlation of the quantity of *B. japonicum* DNA in roots (pg) with the dry weight of soybean nodules (DNW) (g) based on the average values for all the years and combinations of experimental factors. N = 81; Cl—confidence interval; *r*—correlation coefficient; $r^2$—coefficient of determination.

## 4. Discussion

The soybean yield is determined not only by physical, chemical and biological traits of the soil, but, above all, it depends on the of pattern of climate conditions, i.e., air temperature, as well as total rainfall and its distribution [15,38,39]. There was recorded a significant negative effect of low air temperature on the germination and emergence in plants cv. Aldana; hence a reduced plant density and a relatively low air temperature in 2018 and 2019 in July and August, and an accompanying low total rainfall affecting growth, development and yield of soybean. Similar observations were made

before not only in Poland [11] but also in the United States [40], Serbia [41] or in China [14]. The average air temperature at the flowering and pod formation stage, although higher than the long-term mean in Poland, was much lower than 25 °C, which Montanez et al. [8] consider most favorable for the soybean yield and biological N fixation.

In Europe, soybean needs approximately 500 mm of rainfall in the growing season, including at least 300 mm at the flowering and fruit formation stage (BBCH 60–79) [41], while some researchers [15,32] highlight that the seed yield of soybean mostly depends on the total rainfall in May, July and August, as soybean plants assimilate approximately 20% N since the beginning of flowering (BBCH 50), and 80% during generative development (BBCH 60–90) [39]. On the other hand, in our research, at the stage of flowering and pod filling in soybean, the total rainfall was sufficiently high in 2016 and 2017, while in 2018 and 2019 was deeply insufficient. Therefore, the average seed yield in both soybean cultivars was significantly reduced along with a decreasing total rainfall over June–August in subsequent research years, however slightly less intensively in cv. Annushka. Probably also a lower plant density in cv. Aldana resulting from its poorer field emergence under low temperature in 2017 and 2019 was the reason for a lower seed yield than in cv. Annushka.

A shortened maturation due to drought stress contributes to a poorer seed filling, an accelerated senescence, and a reduced seed yield [38]. In our studies with the least favorable rainfall distribution in 2019, the seed yield in cv. Aldana was merely 27.3%, and in cv. Annushka—30.2% of the yield reported in the most favorable year 2016. However, with the average 54.2% plant density, the seed yield in cv. Aldana accounted for up to 71.4%, while the 72.6% protein yield in cv. Annushka, indicating a considerable capacity of soybean plants to compensate for the yield resulting from an excessively low density.

Vyas et al. [42] and Albareda et al. [25] claim that the effect of fertilizing soybean with mineral N is slight but measurable. Mourtzinis et al. [43] observed an increase in yield by 120 kg ha$^{-1}$, La Menza et al. [44]—by up to 11%, as compared with the zero N plots, while Capatana et al. [24] recorded the seed yield up to 30% higher. In the second case, the yield resulted from a higher dry weight of the aboveground soybean and the seed number as well as a higher weight per pod. A high variation in the yield of soybean fertilized with mineral N is related to the plants' response to variability in environmental conditions [42]. In the study carried out in Poland and reported by Lorenc-Kozik and Pisulewska [32], cv. Aldana inoculated with *B. japonicum* yielded 22.7% higher without N fertilization than after applying the rate of 30 kg N. Additionally, Hungria et al. [45] state that inoculation improves seed yield and there is no need for N fertilization in Brazil. Our research showed that in some cases N fertilization in combination with inoculation excels the inoculation alone. According to Capatana [24], an increase in the seed yield in soybean only after inoculation was merely 3.7%, while the mineral fertilizer combined with inoculation increased similarly as mineral N, i.e., about 30%. Albareda et al. [25] observed that the seed inoculation resulted in a significantly higher grain yield and nodulation than in the uninoculated controls. Thus, the research results reported by many authors have confirmed; the effect of mineral N fertilization and a simultaneous seed inoculation on the soybean yield, result in highly diversified findings. In the average of four-year studies, the highest seed yields were recorded after applying HiStick in Annushka and HiStick with 60 kg N·ha$^{-1}$ in both cultivars. Also after using HiStick in Aldana as well as Nitragina plus 60 kg N in both cultivars, seed yield was higher than of others combinations. It must be concluded that under a favorable humidity (2016–2017), a plant response to inoculation and mineral N fertilization was significantly different than in 2018–2019 with a significantly lower total rainfall, whereas almost no quantitatively significant effect of the treatments was observed on the yield of both soybean cultivars. Importantly, neither drought nor high air temperature is favorable for the symbiosis of soybean and *B. japonicum* [11]. Neither is low temperature [9], often observed in Central Europe, as it can have a negative effect on the symbiosis of *Rhizobia* bacteria and it can make the available *Bradyrhizobium* inoculation unsatisfactory [46]. Popovic et al. [23] suggest, however, that mineral N fertilization can reduce the effects of drought, and thus increase the seed yield, especially when environmental conditions improve [42].

In general, the seed yield in soybean depends mainly on the number of pods per plant [39]. Khaledian et al. [29] in their experiment without seed inoculation have found that mineral N at the rates of 30 or 60 kg N·ha$^{-1}$, increased the pod number and the seed number per pod in soybean, however it did not differentiate the 1000 seed weight significantly, while according to Mrkovacki and Morinkovic [47], it also enhanced the height of plants and their aboveground mass. In our research, however, there was found no significant effect of the inoculants and/or mineral N fertilization on the pod number and the 1000 seed weight in the cultivars, yet higher values of both traits were found in cv. Aldana the plant density of which was significantly lower than in cv. Annushka. According to Jarecki and Bobrecka-Jamro [31], the starting N rate of 25 kg N·ha$^{-1}$ facilitates producing significantly more seeds per pod than as a result of Nitragina inoculation, however in our research such results were obtained sporadically in both cultivars.

The applicable literature reports on a possible long-term water stress deteriorating the morphological traits of soybean plants [13,14]. Mrkovacki and Morinkovic [47] claim that mineral N enhances the height of soybean plants. In our research, a favourable effect of mineral N on plant height was confirmed only in cv. Annushka. Additionally, in cv. Annushka significantly higher plants were observed after the application of HiStick + 60 kg N·ha$^{-1}$ than after Nitragina alone, which was not found in cv. Aldana. Moreover, there was recorded a significant effect of seed inoculation and mineral N fertilization and their combination on the HI value, which, according to Tamagno et al. [19], becomes reduced with an increase in BNF, which was not the case for most treatments, although the response of cultivars to the application was quite diversified. On the other hand, the LAI value in most cases was significantly higher in cv. Annushka, especially after seed inoculation and mineral N fertilization at the rate of 60 kg N·ha$^{-1}$ and when combined with seed inoculation.

Soybean uses Ndfa thanks to BFN in the nodules as well as mineral N, which can be derived from the soil (Ndfs) or from the mineral fertilizer (Ndff) after its prospective application. BNF in soybean plants is a sensitive process, dependent mostly on the weather pattern, which, as suggested in our studies, directly affect the development of *B. japonicum* bacteria, nodulation and N fixation. Therefore, probably, both the DNA quantity and the dry weight of nodules decreased significantly each year with a progressively lower total rainfall [48,49]. To verify the presence and the degree of *B. japonicum* proliferation, in our study a molecular technique based on the DNA quantitative analysis has been used [28]. The DNA quantity was considerably enhanced by seed inoculation with HiStick and Nitragina, as well as a combination of HiStick with 30 and 60 kg N·ha$^{-1}$ and Nitragina with 30 kg N·ha$^{-1}$. The results of our molecular analyses in the subsequent research years revealed a significantly lower count of *B. japonicum* in soybean roots and the DNW in the final two research years (2018 and 2019), which recorded a soil drought. Probably a favorable humidity in 2017 increased the amount of bacterial DNA significantly. Jarecki and Bobrecka-Jamro [31], after the application of inoculants, observed an increase in the number of root nodules and dry weight, as well as a higher count of *B. japonicum*. As reported by Furseth et al. [28], or Narożna et al. [50], *Rhizobia* can persist in the soil for many years; hence low amounts of DNA and nodules were found in our studies also on the plots with no inoculant application, which suggests a possibility of *B. japonicum* long-term survival in the soil, in a saprotrophic form, without a host plant. Some authors [20,21] highlight that mineral N fertilization of soybean deteriorates not only the number of nodules but also the seed yield, however it also depends on the N rate. When soybean plants are supplied with mineral N from a nitrogen fertilizer at their disposal, the growth of nodules and the effectiveness of symbiosis decrease [17,18] along with an increase in the mineral N rate [51], which is unfavorable for nodule formation and atmospheric N fixation [11,16]. In our studies in the long-term period, also a significantly limiting effect of mineral N was observed both after the application of 30 and 60 kg N·ha$^{-1}$ on the amount of *B. japonicum* and on the correlation with the dry weight of nodules ($r = 0.6837$). Additionally, Kashuk et al. [21] and Saturno et al. [20] stress that fertilization of soybean with mineral N can negatively affect the number of nodules. Gan et al. [18] and Mrkovacki and Morinkovic [47] report on the low rates of mineral N

being able to increase the number of nodules and their dry weight, which, however, was not noted in our research.

The lack of unequivocal research results on the cultivar response, in particular the clear effect of fertilization and BNF on the yield and some morphological traits of soybean plants, probably results from the climate variability and its impact on the soybean yield variation in different parts of the world [2,52]. The studies by Basal and Szabo [53] carried out in Hungary have shown that drought negatively affects the soybean physiology and yield, regardless of inoculation, however *B. japonicum* seed treatment could be beneficial for soybeans physiology, and consequently, a better yield in moderate drought. Therefore, considering a widespread interest in soybean cultivation in Europe, it seems justifiable, also in Poland, to continue further carefully-controlled field experiments in various regions, mostly in the fields where soybean has never been grown before.

## 5. Conclusions

The thermal and humidity conditions have affected the effectiveness of inoculation and mineral N fertilization and, in addition to the direct impact on the soybean development, they can also modify the yield. A significant decrease in the yield over the years with rainfall deficits shows that soil moisture is an essential factor for an effective N utilization.

The high sensitivity of the nodulation process in soybean to extreme water deficits resulted in a significant reduction in the amount of *B. japonicum* DNA and in the dry weight of nodules.

Despite a more numerous colonization of roots by *B. japonicum* and thus a higher weight of nodules, especially in combinations in which inoculation was applied without mineral N fertilization, a high variability of the results in particular research years did not help providing a definite statement on the effect of a separate application or simultaneous fertilization of soybean with mineral N and the seed inoculation on the morphological traits in soybean plants and the yield.

A high capacity of soybean to compensate the yield when exposed to an excessively low plant density consisted in an increase in the number of pods per plant, seed weight per pod and the 1000 seed weight.

**Author Contributions:** Conceptualization, J.P. and M.B.; data curation, M.B.; formal analysis, A.B.-C. and M.B.; funding acquisition, J.P.; investigation, J.P. and A.B.-C.; methodology, J.P., A.B.-C. and M.B.; project administration, J.P.; resources, J.P. and A.B.-C.; supervision, J.P.; validation, J.P., A.B.-C. and M.B.; visualization, M.B.; writing—original draft, J.P.; writing—review and editing, A.B.-C. All authors have read and agreed to the published version of the manuscript.

**Funding:** This research has been funded by the Polish Ministry of Agriculture and Rural Development, Project: Increasing the use of domestic feed protein for the production of high-quality animal products under sustainable development conditions. Project number HOR 3.6./2016/2020.

**Acknowledgments:** This study was made possible by a grant of the Polish Ministry of Agriculture and Rural Development, Project No. HOR 3.6/2016–2020. The molecular analyses were conducted with use of equipment financed from project 'Stage 2 of the Regional Centre of Innovativeness'—UTP University of Science and Technology in Bydgoszcz.

**Conflicts of Interest:** The authors declare no conflict of interest.

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
