# Peer review of "Response of Soybean (Glycine max (L.) Merrill) to Mineral Nitrogen Fertilization and Bradyrhizobium japonicum Seed Inoculation"

_agronomy, doi:10.3390/agronomy10091300_

Round 1

Reviewer 1 Report

This manuscript has significantly improved since the previous review. However, there is one major issue with the statistical analysis that authors must address before.

The Authors say that this is a two factors experiment while in the methods there are four factors described:  year (four levels: 2016, 2017, 2018, 2019), cultivar (two levels: Annushka, Aldana), inoculation (two levels: Nitragina, HiStick), N fertilization (three levels: 0, 30, 60 kg N ha-1). This article has very good data that must be properly analyzed to answer the research questions proposed by the Authors. Indeed, the Authors do not need to explain all the interactions between the four factors, just the ones that correspond to the research questions but all interactions must be included in the ANOVA.

Here is one article that may help on the statistical analysis:

Moore, K. J., & Dixon, P. M. (2015). Analysis of combined experiments revisited. Agronomy Journal, 107, 763–771

This experiment was performed in the same location for four years. Therefore, each year could be considered as a different environment in the ANOVA. If the statistical analysis array with significant effect (which it is most likely to occur) of the environment (year of the experiment), then, the Authors can explore some correlations of temperature, rainfall, and solar radiation (or their deviation from their historical averages) with the traits differences between treatments. Solar radiation will not change much, but temperature and rainfall will change. Some of these changes are already mentioned by the Authors, but there is not sufficient scientific evidence yet to support it. If there is a significant correlation between temperature or rainfall (in a specific period or the whole cycle) and differences in seed yield between the control treatment and inoculation treatment as an example, then the Authors could scientifically claim the temperature and/o rainfall effect that they saw in the field. Otherwise, it is very difficult to understand these results and make inferences with them.

Here are some revisions:

#14-17: Very important sentence but it is very confusing. The Authors may consider to rephrase it to something similar to “The aim of the study is to evaluate the consistency of seed yield response of two soybean cultivars to three different rates of mineral N fertilization and two seed inoculation treatments with B. japonicum in field conditions over four years regardless of previous B. japonicum presence in the soil.”

Introduction:

#31-42: Authors may consider merging the 1st and 2nd paragraphs.

# 43: “Most of the N” instead of “Most N”

#43-47: Again, the Authors may consider merging the 3rd and 4th paragraphs.

#55: The Authors may consider using the crop development scale of Fehr & Caviness, (1977) as it is the most commonly used in soybean. Replace “BBCH 60-75” by “R1 to R5 stage”.

Fehr, W. R., & Caviness, C. E. (1977). Stages of soybean development. Special Report 80. Iowa Agriculture and Home Economics Experiment Station, Iowa State University, Ames.

#67: “however” is twice in the same line. This sentence should be improved.

#68: “as” is repeated.

#89: Replace “BBCH 49” with the corresponding Fehr and Caviness stage.

#104: Dot is missing between “reasons” and “The”

#106-108: Reduce the brackets to the most important traits and put “eg.”

Materials and Methods:

#112-114: Change accordingly to the improved statistical analysis.

#124-125: Are these two cultivars the most common ones in Poland? Why their selection?

#130-131: This phrase as it is now indicated that Nitraginia and HiStick are the same, then why to test differences between them? The fact that only Nitraginia was available in Poland in the past and now there is one more choice of inoculant (HiStick) is worth to mention it here.

Results:

#182-185: The Authors may consider modifying the first paragraph of results and state the most important findings instead of some experimental constraints.

#188-194: This is great but needs more scientific support. My suggestion above tries to encourage the authors to run some correlation analysis between the average temperatures and rainfall and the treatment responses. Perhaps better if the average of temperatures and rainfall corresponds to crop stages periods instead of months (e.g. R1-R3 or R3-R5).

#212: Modify the table accordingly after improving the ANOVA.

#218-221: Did the Authors tested the differences in seed protein concentration between inoculation and N treatments. There is no mention of it here. If there are no significant differences (from ANOVA) then it is ok to summarize by variety. Still, the year effect must be endorsed by the ANOVA.

Also, the Authors may consider exploring whether there is a negative relationship between seed yield and protein concentration as some literature shows. If yes, is this relationship affected by the cultivar, inoculation, and N fertilizer?

#223: Modify the table accordingly after improving the ANOVA.

Discussion:

#288-298: Phenological stages should be expressed as Fehr and Caviness. The Authors should show some Pearson correlation coefficients and their p values to support the statements in this paragraph.

#306: delete the  “although

#306-308: Good discussion here. Here, these two new articles may be considered by the authors to include in the discussion:

  • Hungria, M., Nogueira, M. A., Campos, L. J. M., Menna, P., Brandi, F., & Ramos, Y. G. Seed pre‐inoculation with Bradyrhizobium as time‐optimizing option for large‐scale soybean cropping systems. Agronomy Journal.
  • La Menza, N. C., Monzon, J. P., Lindquist, J. L., Arkebauer, T. J., Knops, J. M., Unkovich, M., ... & Grassini, P. (2020). Insufficient nitrogen supply from symbiotic fixation reduces seasonal crop growth and nitrogen mobilization to seed in highly productive soybean crops. Plant Cell Environ. https://doi. org/10.1111/pce, 13804.

Hungria et al., (2020) states that inoculation improves seed yield and there is no need for N fertilization in Brazil. However, Table 2 in the Author’s article shows that in some cases N fertilization combination with inoculation excels the inoculation alone. Therefore, when soybeans are pushed into areas with low temperatures in combination with some degree of water stress the inoculation may partially fail and the plant needs other N sources (indigenous soil N, N fertilizer) to satisfies its demand. On the other side, La Menza et al., (2020) show the plant mechanisms of N limitation in high yield irrigated environments in Nebraska, US which is not as cold as Poland. In that study, differences in N uptake occurs up to R5, but the largest differences were between R1 and R3. There may be good to discuss what would happen if the N fertilization of 30 or 60 kg ha-1 in the current study would have been applied between R1 and R3 in combination with inoculation. This may encourage the Authors, and readers as well, to develop some future research ideas in which inoculation is applied but the decision of applying N is based on some predictive factor before or at R1 such as soil water content at R1 for example.

Conclusions:

They are fine but need to modify, if needed, after improvements in the ANOVA.

Author Response

We thank to Reviewer for valuable comments and suggestions that have contributed to the improvement (as we hope) of our manuscript and encouraged us to analyze issues previously not  taken into account.

Modifications in the text of manuscript are highlighted in gray and Author's responses to reviewer's comments are marked as ‘AR’.

Comments and Suggestions for Authors

This manuscript has significantly improved since the previous review. However, there is one major issue with the statistical analysis that authors must address before.

The Authors say that this is a two factors experiment while in the methods there are four factors described:  year (four levels: 2016, 2017, 2018, 2019), cultivar (two levels: Annushka, Aldana), inoculation (two levels: Nitragina, HiStick), N fertilization (three levels: 0, 30, 60 kg N ha-1).

AR: It was assumed to be a two-factor experiment: variety was the first factor, and the method of nitrogen supply / uptake (Inoculation and/or N fertilization) was the second factor. As the years went by, another (third) factor of the experiment arrived - a year, which we did not really specify, although it was included in the analyzes. We fixed it now and we included the year as the third factor.

This article has very good data that must be properly analyzed to answer the research questions proposed by the Authors. Indeed, the Authors do not need to explain all the interactions between the four factors, just the ones that correspond to the research questions but all interactions must be included in the ANOVA.

Here is one article that may help on the statistical analysis:

Moore, K. J., & Dixon, P. M. (2015). Analysis of combined experiments revisited. Agronomy Journal, 107, 763–771

This experiment was performed in the same location for four years. Therefore, each year could be considered as a different environment in the ANOVA. If the statistical analysis array with significant effect (which it is most likely to occur) of the environment (year of the experiment), then, the Authors can explore some correlations of temperature, rainfall, and solar radiation (or their deviation from their historical averages) with the traits differences between treatments. Solar radiation will not change much, but temperature and rainfall will change. Some of these changes are already mentioned by the Authors, but there is not sufficient scientific evidence yet to support it. If there is a significant correlation between temperature or rainfall (in a specific period or the whole cycle) and differences in seed yield between the control treatment and inoculation treatment as an example, then the Authors could scientifically claim the temperature and/o rainfall effect that they saw in the field. Otherwise, it is very difficult to understand these results and make inferences with them.

Here are some revisions:

#14-17: Very important sentence but it is very confusing. The Authors may consider to rephrase it to something similar to “The aim of the study is to evaluate the consistency of seed yield response of two soybean cultivars to three different rates of mineral N fertilization and two seed inoculation treatments with B. japonicum in field conditions over four years regardless of previous B. japonicum presence in the soil.”

AR: Sentence has been rephrased as suggested by the reviewer after a slight modification, as we analyzed various soybean parameters, not only the yield.

Introduction:

#31-42: Authors may consider merging the 1st and 2nd paragraphs.

AR: Paragraphs have been merged

# 43: “Most of the N” instead of “Most N”

AR: Corrected

#43-47: Again, the Authors may consider merging the 3rd and 4th paragraphs.

AR: Paragraphs have been merged

#55: The Authors may consider using the crop development scale of Fehr & Caviness, (1977) as it is the most commonly used in soybean. Replace “BBCH 60-75” by “R1 to R5 stage”.

Fehr, W. R., & Caviness, C. E. (1977). Stages of soybean development. Special Report 80. Iowa Agriculture and Home Economics Experiment Station, Iowa State University, Ames.

AR: We used description for reproductive stages as R2 and R5 in 'Materials and Methods' and we added Fehr and Caviness (1977) to our ‘References’. We did not use the BBCH scale to describe our research. We used BBCH scale in the ‘Introduction’ and the ‘Discussion’ as originally used by the authors cited.

#67: “however” is twice in the same line. This sentence should be improved.

AR: Improved.

#68: “as” is repeated.

AR: Corrected.

#89: Replace “BBCH 49” with the corresponding Fehr and Caviness stage.

AR: Explained above.

#104: Dot is missing between “reasons” and “The”

AR: Corrected.

#106-108: Reduce the brackets to the most important traits and put “eg.”

AR: Corrected.

Materials and Methods:

#112-114: Change accordingly to the improved statistical analysis.

AR: Changed

#124-125: Are these two cultivars the most common ones in Poland? Why their selection?

AR: Aldana is registered in The Polish National list of cultivars and is 000, while Ukrainian Annushka is 0000 and is registered in CCA; we selected these cultivars, because both are the earliest cultivar suitable for cultivation in Poland. We added this information to 2.1.

#130-131: This phrase as it is now indicated that Nitraginia and HiStick are the same, then why to test differences between them? The fact that only Nitraginia was available in Poland in the past and now there is one more choice of inoculant (HiStick) is worth to mention it here.

AR: These are two different products, manufactured by different producers probably (information is not disclosed) with different way and with strains of B. japonicum. In the past, Nitragina was produced by a different manufacturer, so we felt there was no point in mentioning it. Nitragina produced by IUNG-PIB has been available from about the same time as HiStick (BASF).

Results:

#182-185: The Authors may consider modifying the first paragraph of results and state the most important findings instead of some experimental constraints.

AR: First paragraph has been moved. There is right under table 4 now.

#188-194: This is great but needs more scientific support. My suggestion above tries to encourage the authors to run some correlation analysis between the average temperatures and rainfall and the treatment responses. Perhaps better if the average of temperatures and rainfall corresponds to crop stages periods instead of months (e.g. R1-R3 or R3-R5).

AR: We agree it is very interesting, but such detailed analysis will be the subject of a publication that we plan to develop in the near future.

#212: Modify the table accordingly after improving the ANOVA.

AR: Despite considering the year as the third factor, the table did not need to be modified. The C×T×Y interaction result has been added.

#218-221: Did the Authors tested the differences in seed protein concentration between inoculation and N treatments. There is no mention of it here. If there are no significant differences (from ANOVA) then it is ok to summarize by variety. Still, the year effect must be endorsed by the ANOVA.

AR: We added results of suggested analysis.

Also, the Authors may consider exploring whether there is a negative relationship between seed yield and protein concentration as some literature shows. If yes, is this relationship affected by the cultivar, inoculation, and N fertilizer?

AR: We tried to do it. Summarizing the four-year results, the correlation between the seed yield and the protein content in the seeds was not observed. A more in-depth analysis showed that due to the huge variability of atmospheric conditions in subsequent years and the different responses of cultivars to these conditions, it would be a great misuse to generalize that obtaining high yields causes a decrease in protein content, as some of the results concerning one of the cultivars contradict this statement.

#223: Modify the table accordingly after improving the ANOVA.

AR: Table 3 (now Table 4) has been modified.

Discussion:

#288-298: Phenological stages should be expressed as Fehr and Caviness.

The Authors should show some Pearson correlation coefficients and their p values to support the statements in this paragraph.

AR: We believe that we cannot change the nomenclature from the original literature cited.

We analyzed suggested data to support our discussion. They are presented in the new Table 2. Related information is also addend in ‘Materials and Methods’ (2.4).

#306: delete the  “although

AR: Deleted.

#306-308: Good discussion here. Here, these two new articles may be considered by the authors to include in the discussion:

  • Hungria, M., Nogueira, M. A., Campos, L. J. M., Menna, P., Brandi, F., & Ramos, Y. G. Seed pre‐inoculation with Bradyrhizobium as time‐optimizing option for large‐scale soybean cropping systems. Agronomy Journal.
  • La Menza, N. C., Monzon, J. P., Lindquist, J. L., Arkebauer, T. J., Knops, J. M., Unkovich, M., ... & Grassini, P. (2020). Insufficient nitrogen supply from symbiotic fixation reduces seasonal crop growth and nitrogen mobilization to seed in highly productive soybean crops. Plant Cell Environ. https://doi. org/10.1111/pce, 13804.

Hungria et al., (2020) states that inoculation improves seed yield and there is no need for N fertilization in Brazil. However, Table 2 in the Author’s article shows that in some cases N fertilization combination with inoculation excels the inoculation alone. Therefore, when soybeans are pushed into areas with low temperatures in combination with some degree of water stress the inoculation may partially fail and the plant needs other N sources (indigenous soil N, N fertilizer) to satisfies its demand.

On the other side, La Menza et al., (2020) show the plant mechanisms of N limitation in high yield irrigated environments in Nebraska, US which is not as cold as Poland. In that study, differences in N uptake occurs up to R5, but the largest differences were between R1 and R3.

There may be good to discuss what would happen if the N fertilization of 30 or 60 kg ha-1 in the current study would have been applied between R1 and R3 in combination with inoculation. This may encourage the Authors, and readers as well, to develop some future research ideas in which inoculation is applied but the decision of applying N is based on some predictive factor before or at R1 such as soil water content at R1 for example.

AR: We thank the reviewer for a valuable hint. We used the proposed information from the article of Hungria et al. (2020) in the ‘Discussion’ and this articles was included in the reference list. We thought about using the information from the article of La Menza et al. (2020), but we gave up. Due to the recent climate changes, much higher temperatures in recent years in Poland than in the past, we are concerned that the statement that 'Nebraska, US which is not as cold as Poland' could be confusing.

Discussion on the issue what would happen if the N fertilization of 30 or 60 kg ha-1 in the current study would have been applied between R1 and R3 in combination with inoculation could be probably interesting, but it would only be theoretical. We could not say anything specific about it, refer to our own results, because we did not analyze it. ‘

Conclusions:

They are fine but need to modify, if needed, after improvements in the ANOVA.

AR: There is no need  to modify them.

Reviewer 2 Report

Please check the LSD test. If is possible change with t - test or other for multi factors experiment

Author Response

We thank to Reviewer for valuable suggestions that have contributed to the improvement (as we hope) of our manuscript.

Modifications in the text of manuscript are highlighted in gray and Author's responses to reviewer's comments are marked as ‘AR’.

Comments and Suggestions for Authors

Please check the LSD test. If is possible change with t - test or other for multi factors experiment.

AR: Changed

Details highlighted/ indicated in the PDF file (peer-review-8418243.v1.pdf) attached by the reviewer:

#118: Change the term “forecrop”

AR: The term 'forecrop' refers to the spring wheat, grown before soybean each year.

Table 1. Give also the level of Soil Organic matter

AR: The level of soil organic matter was not measured.

#136: 900000.

AR: In description of experiments in Europe we use plant density in m2. In the case of test plots, it is more practical to give the number of seeds per square meter.

# 178 – MANOVA?

AR: ANOVA with repeated measures was used, not MANOVA.

Table 2 - Reviewer’s comments: Maybe tn.ha-1? YEAR interaction effect. CxTxY ? have effect? The LSD test does not used in multi factor experiments. Just for one factor.

AR: Of course, ‘tons’, not ‘kg’.

We changed statistical test to Tukey’s HSD and year interaction effect C×T×Y was measured.

Round 2

Reviewer 1 Report

This is a nice article and data set. The Authors have improved the manuscript to the extent of being publishable. There are some minor details that would like the Authors to address.

#112 The second factor should be “N fertilization and/or seed inoculation” instead of “seed inoculation” as it is stated later in line # 170.

Either here or in line #170, the Authors should make a statement of why they decide to cluster N fertilization and inoculation as one factor instead of two different factors and their interaction between them.

Table 2: It is still confusing the way that the Authors decided to show the ANOVA table. What does the “–“ mean? Using merging cells in this kind of table is not recommended. It is also hard to see that there is no significant  C x T x Y interaction. Authors should make a statement on that an discuss it.

Table 3: it is a nice table. Maybe in future research, the authors could try to average the rainfall between crop stages instead of months. Good job here.

Table 4: Thanks for improving that table, however, it is still hard to interpret it. For example, there are two asterisks or stars in the C which means to me that there is a cultivar effect on seed protein content but in line #229-230 the authors stated that there was no effect of the cultivar on seed protein… This needs to be clarified. I also encourage the authors to edit this table (and table 2 as well) to make it more understandable.

#227-230: Table 4 shows a significant effect of the treatment (T), therefore, the Authors should make a statement here about the effect of inoculation and/or fertilization on seed protein content. Seed protein in soybean is an important trait that has been decreasing over time.

#318 the word “although” should be deleted

Author Response

We thank the reviewer for the patience and for helping us improve our manuscript.

Modifications in the text of manuscript are highlighted in gray and author's responses to reviewer's comments are marked as ‘AR’.

Comments and Suggestions for Authors

This is a nice article and data set. The Authors have improved the manuscript to the extent of being publishable. There are some minor details that would like the Authors to address.

#112 The second factor should be “N fertilization and/or seed inoculation” instead of “seed inoculation” as it is stated later in line # 170.

Either here or in line #170, the Authors should make a statement of why they decide to cluster N fertilization and inoculation as one factor instead of two different factors and their interaction between them.

AR: We introduced suggested changes.

Table 2: It is still confusing the way that the Authors decided to show the ANOVA table. What does the “–“ mean? Using merging cells in this kind of table is not recommended. It is also hard to see that there is no significant  C x T x Y interaction. Authors should make a statement on that an discuss it.

AR: The table has been completed and corrected. Symbol '-' meant the inability to assess the effect. Now it has been replaced by the abbreviation 'i.m.' (impossible to measure) We hope that now this is understandable.

Table 3: it is a nice table. Maybe in future research, the authors could try to average the rainfall between crop stages instead of months. Good job here.

AR: Thank you for your kind words and positive feedback.

Table 4: Thanks for improving that table, however, it is still hard to interpret it. For example, there are two asterisks or stars in the C which means to me that there is a cultivar effect on seed protein content but in line #229-230 the authors stated that there was no effect of the cultivar on seed protein… This needs to be clarified. I also encourage the authors to edit this table (and table 2 as well) to make it more understandable.

AR: Thank you for pointing out our error (# 229-230, now # 230-231). We corrected this. Also Table 4 has been completed, as Table 2 and we hope that now it is more understandable.

#227-230: Table 4 shows a significant effect of the treatment (T), therefore, the Authors should make a statement here about the effect of inoculation and/or fertilization on seed protein content. Seed protein in soybean is an important trait that has been decreasing over time.

AR: Indeed, previously we had added the data to Table 4, however, we wrote nothing about it. We completed this deficiency.

#318 the word “although” should be deleted

AR: Deleted

This manuscript is a resubmission of an earlier submission. The following is a list of the peer review reports and author responses from that submission.

Round 1

Reviewer 1 Report

Review of The Effect of Mineral N Fertilization and Seed, 3 Inoculation on the Yield of Soybean (Glycine max (L.) Merill) under Highly Diversified Thermal and  Humidity Conditions

Line 17-18.  Meaning unclear.  “Yield was getting smaller”

Line 123.  The statistical section needs to be expanded and the experimental design and replication described.  Were treatments rerandomized each year.  Also, how were year effect analyzed in the design?

Line 184.  What is meant by “unfavorable thermal conditions”.  Why would one cultivar be affected more then another if both are adapted to the region.

Line 189.  Here and elsewhere rewrite to clarify meaning.  There is a translation issue here.You don’t need to insert the work “significantly”

Line 189.  I suggest presenting the results in the order of the objectives:  The aim of our research was assessment of the effect of inoculating seeds with preparations containing B. japonicum and mineral N fertilization on the yield, effectiveness of seed inoculation and some morphological plant traits of 2 soybean cultivars  .  in the current form, the results are hard to follow.  Looks like a lot  of data is presented with little thought about what is important to present.

Figure 2.  Redo figure.  Why aren’t the two collection points line up at each year?  i.e., if data is collected in 2017, 2018, and 2019; why isn’t data show for the year?

Figure 3, 4, and 6. 7, 8 and 9.  Are the letters within the figure for mean separation?  Indicate that is footnotes. A continuous line graph  is not the appropriate way to show this data .  A bar graph or a table should be use.  A lot of the data appears to be no significant differences. Does it all need to be presented?

Discussions and conclusions should revolve around the objective points.  It seems the authors spend more time discussing climatic effects than they spend on talking about results  related to the objectives

Reviewer 2 Report

The manuscript is written in high standard. The methodology of the study is correct. The processing of the literature is adequate. There are still possibilities to study of the correlation between results. I suggest deeper exploration of the relationship between data in the future.

Corrections:

The use of units of measurement is not integrated ( 80 kg kg.ha-1; 60 kg N.ha-1, 60 kg N.ha-1;..... . (e.g. in rows 134, 271, 327, 345, 357).

Reviewer 3 Report

This manuscript aims to determine the effect of mineral nitrogen fertilization and seed inoculation on the yield of soybean in Poland. Although this is not a new topic nor novel, I consider that the set of experiments in this article are well done and can be relevant considering that not much research in this topic has been done at the latitude of the study where soybean production is expanding in many other regions of the world. However, the authors claim in the title “…under Highly Diversified Thermal and Humidity Conditions” which can not be supported by the current experimental setting. It is rather an interannual variation in temperature and rainfall. Then, concluding “Thermal and humidity conditions affect the effectiveness of inoculation and mineral N fertilization and determine soybean yield” which ignores the effect of temperature and water in plant growth, and N demand or requirements in soybean. The lack of response in 2018 and 2019 experiments maybe for the simple reason that the maximum possible yield under these harsh weather conditions was low (< 2 Mg ha-1), then low N demand. Therefore, indigenous soil N supply may have been sufficient to cover plant N demand. Years 2016 and 2017 explore a different range of yield and N demand, then treatment responses are seen. 

Here are my comments:

#4-5: Title should be modified. (see explanation above)

Abstract:

#13-16 Divide it into two or three sentences. Statement of the problem, objectives, methods.

Introduction:

It needs to be reorganized. Also, highlight the importance of the current study and why it is different from other previous studies in terms of for example treatment combination never tested before (if it applies), inoculation and fertilization technologies used (have they been tested before), weather conditions and latitude, crop cycle, soil type, yield potential, field without soybean crop and inoculation for more than xx years (if that applies). What is new? Then the specificities can be described in material and methods.

#67: “The ability of soybean plants to feed on mineral N and the one derived from the symbiosis with B. japonicum are independent” Is this a typo? Do the authors mean they are not? Usually, soybean uses indigenous soil N and if it is not enough, they fix N from the air. There is a well know trade-off between these two N supply sources (see in reference #5 Salvagiotti et al., 2008). Another reference could be:

Streeter, J., & Wong, P. P. ( 1988). Inhibition of legume nodule formation and N2 fixation by

nitrate. Critical Reviews in Plant Sciences, 7, 1–23.

#103-108 These sentences would be more useful at the beginning of the introduction. Also, the fact that other countries are trying this may not be enough to justify the research. Why the authors did expect to find something different?

#110-110 “some morphological plant traits” which ones? I suggest adding “(e.g…….)”

Materials and Methods

#114-115 Which are the two factors? Seed inoculation and fertilization? How do authors account for the year variation in the statistical analysis? And for the varieties?

#116-117 This soil class difficult is to understand. I encourage the authors to improve this. Maybe they have data on the percentages of silt, clay, and sand, and organic matter.

#117-118 This is interesting. What was the previous crop or crop rotation? Did the authors sow soybean after soybean (monoculture) in the same spot? This should be clarified here. It also seems to be a decreasing trend in the N-NH4 but not in the N-NO3. Do the authors have any thoughts about this?

#123-128 A lot of information should be added here.

The experimental design should be stated. Was it a complete randomized design, with blocks or a split-plot or other? How many replicates?

Why the selection of these two varieties? Also, why these two inoculants? What is the difference between them to expect different results?

What type of N fertilizer was used? Was it broadcasted or incorporated in the soil previous or at sowing?

Results:

The order in which results are presented does not help with the understanding of the research. General aspects such as yield and the morphological traits may be better to present them first. Then, the authors could move to the DNA in the nodules explanation. Some figures are redundant and may not be needed. An ANOVA table is needed to understand whether there are simple effects or interactions. Then, the authors can show the figures needed to interpret these results.

#186 Figure 1: It would be helpful to have the total amount of rainfall during the growing season or the critical period for soybean (R3-R6) or during the seed felling for every year in the plots. Also, please, spell out the months.

Figure 2 and three do not show the year 2016. The authors must state somewhere why.

#217 Figure 4 could be merged with Figure 3.

#224 Figure 5 needs a lot of work. Are there all the years and treatments? Is the control included there? Did the control have nodules? It is hard to interpret this figure and see a correlation without knowing the origin of the data.

#236-240 It would be great if the authors can report the seed protein concentration. Is there any seed oil concentration data to report?

#252 Figure 7 looks redundant.

Discussion

It looks OK, but I am unable to provide much feedback on this section due to the problems that this article has in the previous sections.

#279-282 I suggest reading this article:

De Bruin, J. L., & Pedersen, P. ( 2009). New and old soybean cultivar responses to plant density and intercepted light. Crop Science, 49, 2225–2232.

Between 20-23 plants per square meter should be enough to produce maximum yields in soybean. May be different in the conditions of the current study.

Conclusion

Should be rephrased and shorten. It may change after the authors run the corresponding ANOVA.